

# Laboratory heat transport experiments reveal grain size and flow velocity dependent local thermal non-equilibrium effects

Haegyeong Lee[1], Manuel Gossler[2], Kai Zosseder[3], Philipp Blum[1], Peter Bayer[4], and Gabriel C. Rau[5]

[1]Institute of Applied Geosciences (AGW), Karlsruhe Institute of Technology (KIT), Karlsruhe, Germany
[2]Stadtwerke München (SWM), Munich, Germany
[3]Chair of Hydrogeology, Technical University of Munich, Munich, Germany
[4]Department of Applied Geosciences, Martin Luther University of Halle-Wittenberg, Halle, Germany
[5]School of Environmental and Life Sciences, The University of Newcastle, Callaghan, Australia

**Correspondence:** Haegyeong Lee (haegyeong.lee@kit.edu)

**Abstract.** Heat transport in porous media is crucial for gaining earth science process understanding and engineering applications such as geothermal system design. While heat transport models are commonly simplified by assuming local thermal equilibrium (LTE, solid and fluid phases are averaged), local thermal non-equilibrium (LTNE, solid and fluid phases are considered separately) heat transport has long been hypothesized and reports have emerged. However, experiments with realistic

grain sizes and flow conditions are still lacking in the literature. To detect LTNE effects, we conducted comprehensive laboratory heat transport experiments at Darcy velocities ranging from 3 to 23 $\mathrm{m\,d^{-1}}$ and measured the temperature of fluid and solid phases separately for glass spheres with diameters of 5, 10, 15, 20, 25 and 30 mm. Four replicas of each size were embedded at discrete distances along the flow path in small glass beads to stabilize the flow field. Our sensors were meticulously calibrated and measurements were post-processed to reveal LTNE, expressed as the difference between solid and fluid temperature during

the passing of a thermal step input. To gain insight into the heat transport properties and processes, we simulated our experimental results in 1D using commonly accepted analytical solutions for LTE and a numerical solution of LTNE equations. Our results demonstrate significant LTNE effects with increasing grain size and water flow velocity. Surprisingly, some temperature differences were negative indicating that the heat front propagates non-uniformly likely caused by spatial variations of the flow field. The fluid temperature modeled by the LTE analytical solution exhibited relatively good agreement with experimental

fluid temperature only for grain sizes from 5 mm to 15 mm. However, for larger grain sizes (between 20 mm and 30 mm), the temperature difference between fluid and solid phases became too significant to be represented by an LTE model. Additionally, for larger grain sizes ($\geq 20\,\mathrm{mm}$), the LTNE model failed to predict the magnitude of LTNE for all tested flow velocities due to experimental conditions being inadequately represented by the 1D model with ideal step input. Future studies should employ more sophisticated numerical models to examine the heat transport processes and accurately analyze LTNE effects, considering

non-uniform flow effects and multi-dimensional solution. This is essential to determine the validity limits of LTE conditions for heat transport in natural systems such as gravel aquifers with grain sizes larger than 20 mm.



## 1 Introduction

Accurately describing heat transport in porous media has long been a focus in both engineering and science. In engineering applications, the study of heat transport through porous media is vital for enhancing the design of systems such as chemical
reactors filled with catalysts (e.g., Levec and Carbonell, 1985a), or pebble bed reactors filled with coolants (e.g., Novak et al., 2021). Understanding how heat propagates through sedimentary aquifers is also crucial for modeling thermal responses and designing sustainable geothermal systems (e.g., Vafai, 2005; Banks, 2015; Pophillat et al., 2020a, b). Moreover, natural heat propagation serves as a valuable tracer for characterizing streambed thermal properties and water fluxes between groundwater and surface waters (e.g., Rau et al., 2014). A thorough grasp of heat transport across various domains plays a pivotal role in
advancing both scientific knowledge and engineering applications.

When describing heat transport in saturated porous media, two distinct approaches are commonly considered. The most detailed and accurate method involves formulating two differential equations to account for the two-phase nature (i.e., liquid and solid) of heat transport. This approach separates heat flow in the fluid and solid phases into two energy equations, enabling the representation of temperature differences between the two phases. This method is termed the *local thermal non-equilibrium*
(LTNE) approach (Schumann, 1929; Levec and Carbonell, 1985a; Kaviany, 1995; Hamidi et al., 2019). Heat transfer between the phases is depicted by a heat transfer term, comprising a heat transfer coefficient — defined as the ratio of heat exchange between the two phases for a single particle — and a specific surface area, representing the total contact surface area of the porous media (Kaviany, 1995).

An alternative approach involves simplifying the description by volume averaging across the phases of porous media within
a representative elementary volume (REV) (Bear, 1961), resulting in a single energy equation. This method assumes that both the solid and liquid phases are always at the same temperature at their interface and is hence referred to as the *local thermal equilibrium* (LTE) model (de Marsily, 1986; Whitaker, 1991). By disregarding the heat transfer mechanism between the phases, this approach does not distinguish between heat fluxes of fluid and solid phases. It has become the de facto standard model utilized in geoscience literature(de Marsily, 1986).

The first investigations of LTNE transport and conditions were conducted in the field of mechanical and chemical engineering. Levec and Carbonell (1985a, b) reported discrepancies between temperature responses of fluid and solid phases over time from experiments, indicating LTNE effects. They introduced a spatially averaged heat transport model which enabled validation of experimental results under LTNE conditions. Amiri and Vafai (1994) addressed the validity of LTE model demonstrating that LTE becomes not applicable as the particle Reynolds number $Re_p$ (particle's relative velocity with respect to the sur-
rounding fluid) and the Darcy number $Da$ ($Da = \frac{K}{L^2}$; where $K$ is the permeability of porous media and $L$ is the characteristic macroscopic length) increase. Kim and Jang (2002) proposed a criterion for the LTE assumption considering effects of $Da$, Prandtl number $Pr$ (ratio of momentum diffusivity to thermal diffusivity) and the Reynolds number $Re$ (ratio between inertial and viscous forces). Although numerous studies focus on the validity of LTE in relation to important engineering parameters (e.g., $Re$, $Da$, $Pr$, etc.), LTNE studies are increasingly focused on incorporating the detailed physics into the heat transport
model (Pati et al., 2022; Heinze, 2024).





In the field of geosciences, the LTE approach has been widely adopted as a standard practice, often without thorough consideration of the physical field conditions (e.g., Rau et al., 2014; Pastore et al., 2016; Gossler et al., 2019). While previous studies demonstrate the existence of LTNE effects in flow through natural porous media (e.g., Levec and Carbonell, 1985b; Baek et al., 2022; Bandai et al., 2023; Heinze, 2024), there is a noticeable absence of experimental data concerning the relationship between LTNE effects, flow velocity and grain size. Such data could significantly contribute to efforts aimed at establishing the validity conditions for LTE heat transport.

The absence of experimental data representative of real-world conditions has spurred theoretical examinations of LTNE and its potential impact. Gossler et al. (2020) undertook a theoretical investigation to elucidate how LTNE effects evolve with grain size and flow velocity, employing the two-equation model (LTNE model). Their study uncovered a knowledge gap regarding the heat transfer coefficient. To address this, they compiled experimental data from mechanical engineering to derive an empirical relationship, subsequently employing it to delineate LTNE conditions. Their findings indicated that LTNE conditions, characterized by a difference between solid and fluid temperatures, become significant for grain sizes $> 7$ mm and flow velocities $> 1.6$ m d$^{-1}$ (Gossler et al., 2020). However, their results await validation. Experiments conducted by Baek et al. (2022) revealed that LTNE can occur even for smaller grain size (0.76 mm) and fast flow velocities $> 20$ m d$^{-1}$. Bandai et al. (2023) detected the temperature difference between fluid and solid phases in heat transport experiments as the signature of LTNE effects and compared the experimental to a numerical model. Also, they illustrated that the magnitude of temperature difference between two phases grows as Darcy velocity and effective thermal conductivity of fluid increase, representing sensitive parameters in the LTNE model.

We investigate the presence of local thermal non-equilibrium (LTNE) effects during heat flow in porous media. In this study, we present findings from an advanced laboratory experiment on heat transport. Comprehensive measurements of temperature responses in both fluid and solid phases are performed, varying grain sizes (5 - 30 mm) and flow velocities (3 - 23 m d$^{-1}$) in response to step-like temperature changes. Our analysis aims to elucidate the influence of grain size and flow velocity on heat transport in porous media.

## 2 Material and methods

### 2.1 Experimental setup and measurements

We employed specialized experimental instrumentation developed by Gossler et al. (2019) in a preceding study on heat transport. Our adapted experimental configuration comprises an acrylic glass column with a length of 1.5 m and an inner diameter of 0.29 m, a refrigerated bath circulator (WCR-P22, Witeg Labortechnik GmbH, Germany), an eight-channel peristaltic pump (Ismatec Ecoline, Kinesis Australia Pty Ltd, Australia), and an outflow tank. The schematic representation of the utilized apparatus is shown in Fig. 1.

Temperature time series during the heat transport experiments were measured by 2 types of four-wire Pt100 sensors. One type, referred to hereafter as Pt100 type A, was a hermetically sealed resistance temperature detector with diameter of 2 mm and an approximate resolution of $\pm$ 0.01 °C, which was used to measure the temperature of fluid and solid phases (Fig.



1b). The other type, referred to hereafter as Pt100 type B, was sheathed with a length of 18 cm and diameter of 3 mm (Fig.
90  1d). It featured an accuracy of ± 0.03 °C and was used for revealing boundary conditions. The temperature sensors were
electronically controlled by 20 data acquisition modules, Pt104A (Omega Engineering Inc., USA) each having 4 channels at
1 second intervals (1 Hz measurement frequency), which is shown in Fig. 1a. The temperature response time of these devices
was measured at approximately 4.7 seconds.

A total of 24 special LTNE probes were hand-crafted for 6 different glass sphere sizes (4 for each diameter of 5 mm,
10 mm, 15 mm, 20 mm, 25 mm and 30 mm) to separately measure the temperature in the solid phase (at the center of the
sphere) and at both sides of the surrounding fluid phase as shown in Figure 1b. For the solid phase measurement, each glass
sphere was designed to place a Pt100 type A into the center of the sphere. Each glass sphere with a customized 2.5 mm hole
was manufactured to be used as a grain in the experiments. Temperature sensors were carefully inserted and embedded using
thermally conductive glue with a thermal conductivity of 1.1 W m$^{-1}$K$^{-1}$ minimizing heat transport influences. For the fluid
phase measurements, two temperature sensors were symmetrically placed next to each sphere about 2 mm distance from the
surface (Fig. 1b). Four replicas of each same-sized LTNE probe were fixed on a PVC frame with thickness of 5 mm (Fig. 1c)
and placed at the specific depth in the column for one specific sphere size (Fig. 1d). These LTNE probes measured temperature
development in time series. To determine LTNE effects, heat transport detected by a single probe-unit (Fig. 1b) was considered
as one experiment.

To stabilize the flow field surrounding the glass spheres we embedded them in a bulk consisting of water saturated small
diameter glass beads (1 mm diameter) as otherwise fluid flow would be very sensitive to fluid dynamics or changes in density
caused by the thermal front (i.e., free convection). This decision was based on experience with previous experimentation where
a non-uniform flow field and associated anomalies challenged analysis of transport parameters using temperature measurements
(Rau et al., 2012a, b; Gossler et al., 2019). This is also justified as it was demonstrated that LTNE should be negligible for
grain diameters smaller than 7 mm (Gossler et al., 2020).

The small glass beads were filled above a perforated plate wrapped by filter fleece while the column was vertically positioned.
The glass beads were manually packed in layers of 5-10 cm. Based on our experimental design, temperature sensors were
located through new holes at the column to monitor the temperature breakthrough within the porous medium. During the
packing, the hand-crafted LTNE probes were inserted within the porous media at different depths (i.e., distance along the flow
path) of the column (Fig. 1d):

1. Four 5 mm diameter spheres at 25 cm depth;

2. Four 10 mm diameter spheres at 45 cm depth;

3. Four 15 mm diameter spheres at 65 cm depth;

4. Four 20 mm diameter spheres at 85 cm depth;

5. Four 25 mm diameter spheres at 105 cm depth;

6. Four 30 mm diameter spheres at 125 cm depth.



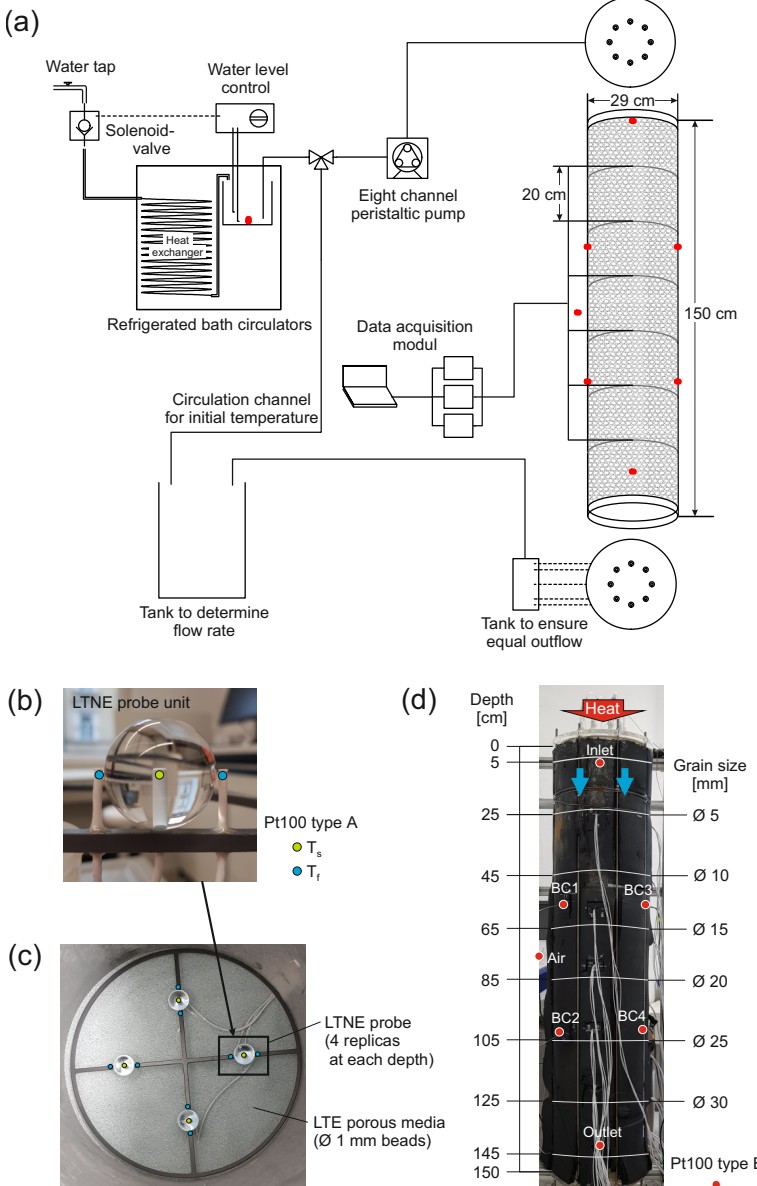

**Figure 1.** Overview of the experimental setup and its details: (a) Conceptual diagram of the flow through experiment, (b) LTNE probe unit; design of the LTNE probe showing one temperature sensor embedded within a sphere measuring the solid phase as well as two sensors on each side measuring the fluid phase, (c) four replicas with the same grain size fixed on the PVC frame; arrangement of the hand-crafted LTNE probes with four replicas for a specific sphere size, consisting of 8 fluid temperature sensors and 4 solid temperature sensors for a specific depth, (d) setup of the column filled with porous media and the LTNE probe arrangements for 6 different depths corresponding to 6 different grain sizes.



The temperature was measured at the top (6 cm depth) and bottom (135 cm depth) inside the column, wall boundary, air temperature, inlet and outlet water temperature to monitor boundary conditions (Fig. 1d).

All temperature sensors underwent calibration within a water-filled bath placed inside the thermostat bath. Various temperature settings (5, 15, 20, 35 °C) were employed, and recordings were taken upon reaching the targeted temperature, ensuring the sensors had equilibrated. To establish a uniform initial temperature across the entire column, water circulation with outlet water was employed. This process facilitated the equilibration of the temperature of porous media and fluid within the pores with the air temperature in the laboratory.

Upon achieving an initial temperature within the range of 24 to 30 °C through circulation, inflow commenced by switching a valve from the circulation channel to the inflow channel. The inflow, sourced from the laboratory tap, was preheated through a heat exchanger within the refrigerated bath, maintaining a temperature between 26-34. The temperature of water in the bath was 5-8 °C higher than the initial temperature, which represents the equilibrated temperature of the system before the heat input was injected. Experimentation concluded when the temperature of all sensors reached a constant value at the culmination of the temperature rise.

Following the insights from Gossler et al. (2019) and their comprehensive testing of various column settings, we adopted the approach of conducting experiments in a vertically oriented column with a step heat input. This configuration yielded unbiased results by minimizing interference from free convection and guided our heat transport investigations. In our experimental setup, both water flow and temperature step input were introduced from the top to the bottom of the vertically positioned column. The heat input mechanism involved the injection of warm water from the top using a peristaltic pump, ensuring a constant flow rate and, consequently, a consistent Darcy flux within the column ranging from 3 to 23 m d$^{-1}$. Subsequently, the outflow was discharged through tubes from the outflow tank connected to the bottom of the column (Fig. 1a). Flow rate quantification was achieved by weighing the collected outflow water on a minute-by-minute basis for each experiment.

The total porosity of the porous media was determined experimentally. Glass beads, comprising the porous medium, were loaded and compacted into a cylinder with an inner diameter of 9.6 cm and a height of 12 cm to measure the weight of the beads. Utilizing the obtained weight from 5 times repeated measurements, cylinder volume and the known density of the glass, the total porosity was subsequently calculated from each measurement and then averaged resulting in 0.37. To ascertain the thermal conductivity and volumetric heat capacity of the glass (solid phase), the *Transient Plane Source* method (TPS) was employed using a HotDisk instrument (TPS1500, C3 Prozess- und Analysentechnik, Germany). The measurements were conducted with the assistance of data acquisition software (Hot Disk Thermal Constants Analyser 7.4.17). The measurement uncertainty of the solid thermal conductivity $\lambda_s$ and the solid volumetric heat capacity $\rho_s c_s$ was 2 % and 7 %, respectively. The physical properties of both the fluid and solid phases are summarized in Table 1.

## 2.2 One-phase model of heat transport in porous media

To describe heat transport during flow through porous media the one-phase advection-diffusion heat transport equation is generally used in hydrogeological applications (Heinze, 2024). This assumes that the temperature at the interface between solid and fluid is always in equilibrium, hence it is termed the *Local Thermal Equilibrium* (LTE) model (Whitaker, 1991). The





**Table 1.** Summary of parameter values of the porous medium, obtained from measurements or literature.

| Parameter | Value | Unit | Source |
|---|---|---|---|
| Initial temperature $T_0$ | 24.0 - 27.5 | $^\circ$C | Measured |
| Temperature input $T_1$ | 29.8 - 37.2 | $^\circ$C | Measured |
| Total porosity $n_t$ | 0.37 | - | Measured |
| Thermal conductivity of fluid (24 $^\circ$C) $\lambda_f$ | 0.6 | W m$^{-1}$K$^{-1}$ | Wagner and Pruß (2002) |
| Thermal conductivity of solid $\lambda_s$ | 1.0 | W m$^{-1}$K$^{-1}$ | Measured |
| Specific heat capacity of fluid (24 $^\circ$C) $c_f$ | 4181.8 | J kg$^{-1}$K$^{-1}$ | Wagner and Pruß (2002) |
| Specific heat capacity of solid $c_s$ | 759.4 | J kg$^{-1}$K$^{-1}$ | Measured |
| Density of fluid (24 $^\circ$C) $\rho_f$ | 997.3 | kg m$^{-3}$ | Wagner and Pruß (2002) |
| Density of solid $\rho_s$ | 2585.0 | kg m$^{-3}$ | Vendor |

equation is as follows (de Marsily, 1986)

$$\frac{\partial T}{\partial t} = D\frac{\partial^2 T}{\partial x^2} - v\frac{\partial T}{\partial x} \tag{1}$$

where $T$ is the temperature of the bulk porous medium ($^\circ$C or K), $t$ is the time (s). The thermal dispersion coefficient $D$ (m$^2$ s$^{-1}$) is defined as (Rau et al., 2012a; Gossler et al., 2020)

$$D = n\left(\frac{\lambda_f}{\rho_b c_b} + \beta\left(\frac{\rho_f c_f}{\rho_b c_b}q\right)^2\right) + \frac{(1-n)\lambda_s}{\rho_b c_b}. \tag{2}$$

The thermal conductivity of the saturated porous media is estimated by arithmetic mean model as a mixing law model (Stauffer et al., 2013; Menberg et al., 2013; Tatar et al., 2021). This model leads to the maxium value of the thermal conductivity for glass packs, which is defined as follow

$$\lambda_b = n\lambda_f + (1-n)\lambda_s, \tag{3}$$

where $n$ is the total porosity; $\lambda_f$ and $\lambda_s$ are the thermal conductivity of the fluid and solid, respectively. Further, $\rho_b$ is the density and $c_b$ is the specific heat capacity of the water saturated porous media (bulk) which, when combined, represent the bulk volumetric heat capacity as (Buntebarth and Schopper, 1998)

$$\rho_b c_b = n\rho_f c_f + (1-n)\rho_s c_s. \tag{4}$$

The fluid and solid densities are $\rho_f$ and $\rho_s$ (kg m$^{-3}$), respectively; $c_f$ and $c_s$ are the specific heat capacity of the fluid and solid (J kg$^{-1}$K$^{-1}$), respectively. The thermal front velocity $v$ is (Rau et al., 2012a)

$$v = q\frac{\rho_f c_f}{\rho_b c_b}, \tag{5}$$



where $q$ is the Darcy velocity (m s$^{-1}$).

The LTE model (Eq. (1)) was solved by an analytical solution as follows (van Genuchten and Alves, 1982)

$$T = \frac{1}{2} erfc \left( \frac{x-v}{2\sqrt{Dt}} \right) + \frac{1}{2} exp \left( \frac{vx}{D} \right) erfc \left( \frac{x+vt}{2\sqrt{Dt}} \right), \tag{6}$$

with the following initial and boundary conditions:

$$T = T_0 \quad \text{at all } x \text{ and } t = 0, \tag{7}$$

$$T = T_1 \quad \text{at } x = 0 \text{ and } t > 0, \tag{8}$$

$$T = T_0 \quad \text{at } x = \infty \text{ and } t > 0. \tag{9}$$

170    Here, $T_0$ is the initial temperature (K) and $T_1$ is the temperature (K) of heat input at the top boundary (x = 0).

Equation (1) simplifies the heat transport description by considering the thermal energy in the porous medium as a bulk. This means it represents a volume averaged temperature as is reflected by the volume averaging of the thermal properties (Eq. (2)-(4)). We note that the dispersion coefficient $D$ in this model incorporates both thermal diffusion through the two phases as well as and hydrodynamic dispersion resulting from the flow through tortuous flow paths. Experiments have demonstrated this

175    to have a non-linear relationship with the flow velocity (Metzger et al., 2004; Molina-Giraldo et al., 2011; Rau et al., 2012a).

### 2.3    Two-phase model of heat transport in porous media

A more precise description follows from separating the temperature in the fluid and solid phases and considering heat transfer between the phases (Amiri and Vafai, 1994). This approach is termed *Local Thermal Non-Equilibrium* (LTNE). The fluid phase (subscript $f$) can be described as (Levec and Carbonell, 1985a; Kaviany, 1995)

$$n\rho_f c_f \frac{\partial T_f}{\partial t} + \rho_f c_f v \frac{\partial T_f}{\partial x} = n\lambda_{f,eff} \frac{\partial^2 T_f}{\partial x^2} + h_{sf} a_{sf}(T_s - T_f), \tag{10}$$

whereas the solid phase (subscript $s$) is described by

$$(1-n)\rho_s c_s \frac{\partial T_s}{\partial t} = (1-n)\lambda_{s,eff} \frac{\partial^2 T_s}{\partial x^2} - h_{sf} a_{sf}(T_s - T_f). \tag{11}$$

180    Here, $T_f$ and $T_s$ are the separate temperatures of the solid and fluid phases, respectively. $\lambda_{f,eff}$ and $\lambda_{s,eff}$ are effective thermal conductivities of the fluid and solid phases, which describe the thermal conductivity of each phase under conduction



conditions without advection (Bandai et al., 2023). These two energy equations are coupled by heat transfer between fluid and solid driven by the temperature difference between the solid and fluid phase and determined by the heat transfer coefficient $h_{sf}$ (W m$^{-2}$K$^{-1}$) as well as the specific surface area $a_{sf}$ (m$^2$). The heat transfer coefficient $h_{sf}$ is the heat exchange across the surface area between the liquid and solid phase $a_{sf}$ (m$^2$), and these are defined as follows (Gossler et al., 2020)

$$h_{sf} = \frac{Nu\lambda_f}{d_p}, \tag{12}$$

$$a_{sf} = \frac{6(1-n)}{d_p}. \tag{13}$$

where $Nu$ is the $Nusselt$ number; $d_p$ is particle (grain) size. The $Nusselt$ number is a dimensionless parameter presenting correlation between the heat transfer coefficient and hydraulic parameters. The correlation proposed by Wakao et al. (1979) is commonly utilized to estimate the heat transfer coefficient, which is derived from experiments in mechanical engineering (Kaviany, 1995; Amiri and Vafai, 1994; Bandai et al., 2023). However, Gossler et al. (2020) suggested another correlation based on an adaptation of the $Nusselt$ number to groundwater conditions by keeping the $Prandtl$ number, a dimensionless parameter in the correlation of Wakao et al. (1979), constant for water at a fixed temperature. Although the correlation of Gossler et al. (2020) is experimentally not validated in porous aquifer conditions, it provides an estimation relevant to shallow groundwater flow regimes. Thus, the present study estimated the heat transfer coefficient by using the correlation of Gossler et al. (2020) and by fitting the LTNE model to the temperature difference between two phases from experimental data to achieve the best model for LTNE effects. The estimation of the heat transfer coefficient with the correlation with the $Nusselt$ number $Nu$ and the $Reynolds$ number $Re$ was performed with the following equations (Gossler et al., 2020):

$$Nu = 1 + 3.1Re^{0.57}, \tag{14}$$

$$Re = \frac{\rho_f\,(q/n)\,d_p}{\mu}. \tag{15}$$

Here, $\mu$ is dynamic viscosity (kg m$^{-1}$s$^{-1}$). And $d_p$ is diameter of a grain (m).

The LTNE model (Eq. (10) and (11)) was solved in a one-dimensional space using *FEniCS* in *Python* (Alnaes et al., 2015). The model domain spans 1.5 m $L$ to represent the experimental setup used in our work. The equations are solved using the finite element method with following initial and boundary conditions (Bandai et al., 2023):

$$T_s = T_f = T_0 \quad \text{for all } x \text{ and } t = 0, \tag{16}$$

$$T_s = T_f = T_1 \quad \text{on } x = 0 \text{ and } t > 0, \tag{17}$$

$$T_s = T_f = T_0 \quad \text{on } x = L \text{ and } t > 0. \tag{18}$$



Spatial and time discretisations were set at 0.5 mm and 1 second, respectively. Thermal breakthrough curves (BTCs) are generated for discrete distances of 0.2, 0.4, 0.6, 0.8, 1.0, and 1.2 m, corresponding to temperature measurement points in the experimental setup for each grain size (Fig. 1d).

Equations (10) and (11) describe the heat flux for fluid and solid phases respectively, allowing temperature difference between the two phases. Accordingly, effective thermal conductivity of each phase is considered in each energy equation to describe thermal conduction and dispersion phenomena (Amiri and Vafai, 1994; Bandai et al., 2023). The effective thermal conductivity $\lambda_{f,eff}$ includes thermal diffusion in the fluid phase and hydrodynamic dispersion in relation to the flow velocity (Rau et al., 2012a). To estimate $\lambda_{f,eff}$, the LTNE model was fitted to the experimental data. The effective thermal conductivity
of the solid $\lambda_{s,eff}$ was considered the same as the thermal conductivity of the solid $\lambda_s$, since thermal conduction of the solid phase is considered unaffected by the flow through.

### 2.4    Analysis of the experimental temperature measurements

To reveal possible LTNE heat transport effects, the temperature difference between the fluid and solid phases over time was calculated based on thermal BTCs for each LTNE probe. The temperature difference between the two phases was computed by
subtracting solid phase temperature from the adjacent fluid phase temperature, since heat transport was stimulated by inflow of heated water. The calculated temperature difference time series is referred to hereafter as $\Delta T(t)$, and values deviating from zero indicate temperature differences between fluid and solid indicating LTNE effects.

Although care was taken for each experiment to commence after thermal equilibration to the initial temperature within the column, slight variations in initial temperatures were observed among the sensors. The temperature difference between a pair
of sensors within each LTNE probe unit was 0.05 K on average. This discrepancy could stem from sensor drift or calibration errors in the intercepts of the calibration curves. Since these discrepancies can obscure LTNE effects, a special data correction procedure was applied to all BTCs. The beginnings and tails of the breakthrough curves (BTCs) were adjusted to mitigate calibration errors of the sensors, making the plausible assumption that the initial and final temperatures were the same for each LTNE probe. The following steps were applied:

–    The temperature records of the fluid and solid phases were normalized for each sensor in a time series by subtracting the initial temperature and being divided by the final temperature (equilibrated temperature at the tails of the BTCs) from the temperature measurement. The result of this is an up- or downward shift of the entire time series.

         –    The normalized temperature of both phases are multiplied by the averaged final temperature of the maximum temperatures in the BTCs of sensors for a single LTNE probe (two fluid temperature and one solid temperature measurements).

This procedure allows evaluation of an improved $\Delta T(t)$ that is consistent and simple to interpret.

We further applied models to describe our experimental observations, assuming both LTE (Eq. (1)) and LTNE (Eq. (10) and (11)) conditions. Here, the temperature measurements from the four probe replicas at the same depth (i.e., eight fluid temperature and four solid temperature measurements as shown in Fig. 1c) were averaged at each time step to represent fluid





and solid temperature for each discrete distance along the flow path. The averaged temperature of fluid and solid phases allows
data analysis with one-dimensional LTE and LTNE models.

## 3   Results

### 3.1   Solid and fluid temperature responses to inflow of heated water

Heat transport experiments revealed evidence of LTNE effects stemming from distinct thermal breakthrough curves (BTCs) for
the solid and fluid phases over time. Figure 2 displays selected BTCs recorded within and next to a sphere for six different grain
sizes, in response to a temperature step input with Darcy flux values of $17.2 \, \mathrm{m \, d^{-1}}$ and $22.8 \, \mathrm{m \, d^{-1}}$. These BTCs, representing
solid and fluid phases, are arranged according to increasing grain diameter, reflecting the expected behaviour of heat transport:
delayed arrival times for the solid phase and increased dispersion over distance.

    A noticeable divergence between the fluid and solid BTCs becomes apparent with larger grain sizes, indicating temperature
discrepancies between the two phases. The experimental procedure, necessitating the replenishment of the water bath with tap
water during the experiment, is evident in the declining tails of the BTCs. However, as this replenishment occurred after the
fluid and solid phases had equilibrated, it was deemed non-influential in our analysis.

### 3.2   Adjusted temperature breakthrough curves

The processed temperature data, based on measurements, is depicted in Fig. 3. This figure showcases temperature values from
sensors at identical depths (Fig. 3a, c, e) and the averaged temperature for both fluid and solid phases at those specific depths
(Fig. 3b, d, f), considering a Darcy flux of $22.8 \, \mathrm{m \, d^{-1}}$.

    As a result of the data post-processing, the thermal breakthrough curves (BTCs) exhibit temperature rise from a consistent
initial temperature, which is induced by heat input. Furthermore, the tails of the BTCs reach a uniform final equilibrated
temperature for sensors at the specific depth corresponding to a particular grain size $d_p$. However, despite the identical flow
velocity, the BTCs of each phase at the same depth display non-alignment due to varying thermal velocities, which depend on
the transversal position of the LTNE probe (Fig. 3a, c, e). Consequently, averaging the temperatures for fluid and solid phases
is necessary to obtain a representative temperature response for each phase at a given grain size and depth (Fig. 3b, d, f).

    In Fig. 3, the BTCs with the averaged solid temperature illustrate deviation from the averaged fluid temperature for the same
grain size $d_p$, consistent with the findings from single temperature measurements of solid and fluid phases in an LTNE probe.

### 3.3   Temperature differences between phases

The temperature contrast between solid and fluid phases, as indicated by adjusted thermal breakthrough curves (BTCs), unveils
the impact of varying grain size and flow velocities on the extent of LTNE effects. In Fig. 4, BTCs for both fluid and solid
phases, along with their corresponding LTNE effects ($\Delta T(t)$), are demonstrated for each grain size at the highest tested Darcy
velocity of $23 \, \mathrm{m \, d^{-1}}$. This example highlights the maximum $\Delta T(t)$ observed among pairs of fluid and solid measurements



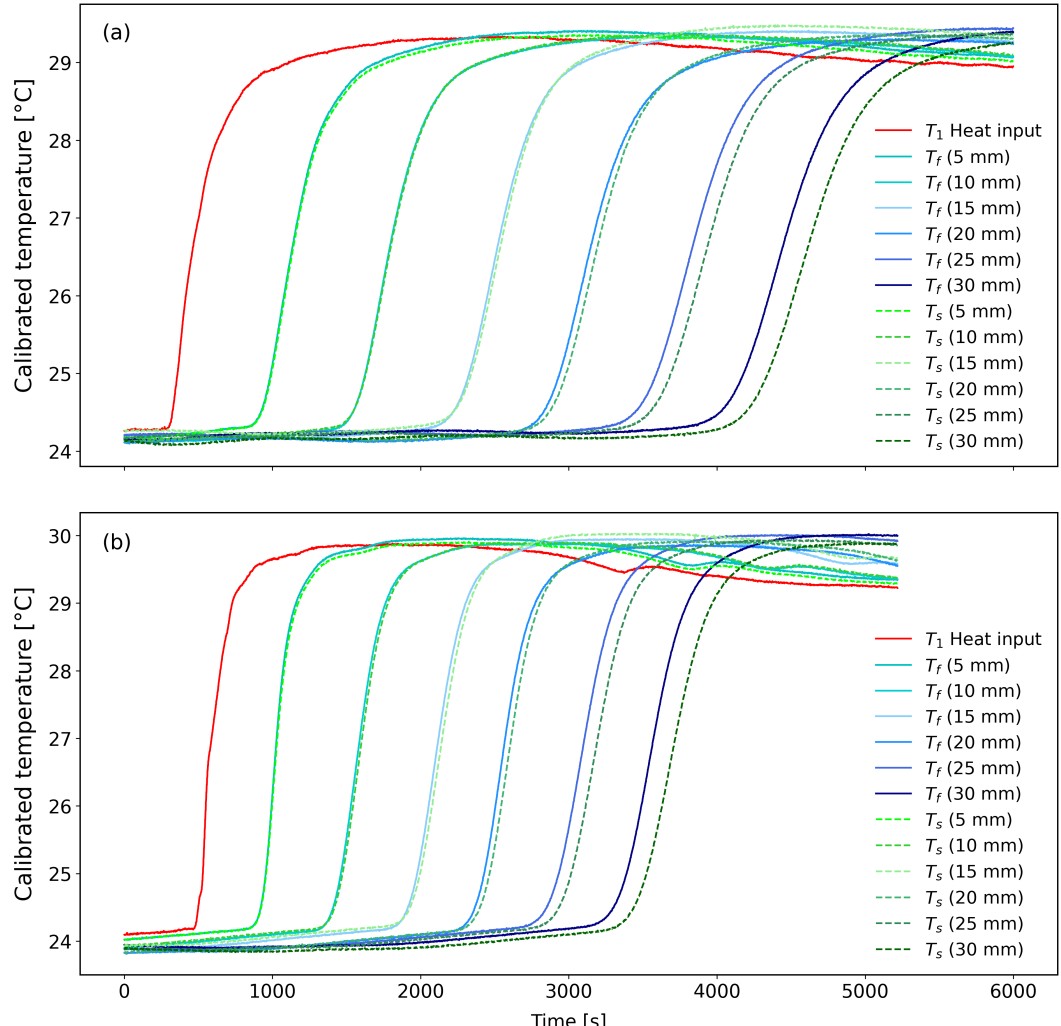

**Figure 2.** Calibrated temperature data yielded thermal breakthrough curves (BTCs) for both fluid and solid phases across six distinct grain sizes in heat transport experiments. The red solid lines present temperature measurements at the top of the column, indicating temperature of heat input into the porous media. (a) The BTCs corresponding to a Darcy velocity of $17.2 \, \mathrm{m \, d^{-1}}$ exhibit variations in temperature between their initial and final states, as depicted in the plotted calibrated temperature measurements. (b) Conversely, the BTCs associated with a Darcy velocity of $22.8 \, \mathrm{m \, d^{-1}}$ illustrate a quicker attainment of equilibrium with the final temperature compared to those reflecting slower Darcy velocities.

for the same grain size. The disparity between fluid and solid BTCs signifies a delayed response in the solid phase, distinctly revealing the LTNE effect.

The results showcase an augmentation in the maximum $\Delta T(t)$, reflecting an amplification of the LTNE effect with increasing grain size. Nevertheless, for grain sizes ranging between $5 \, \mathrm{mm}$ and $15 \, \mathrm{mm}$, an 'inverse pulse' of $\Delta T(t)$ was observed in some






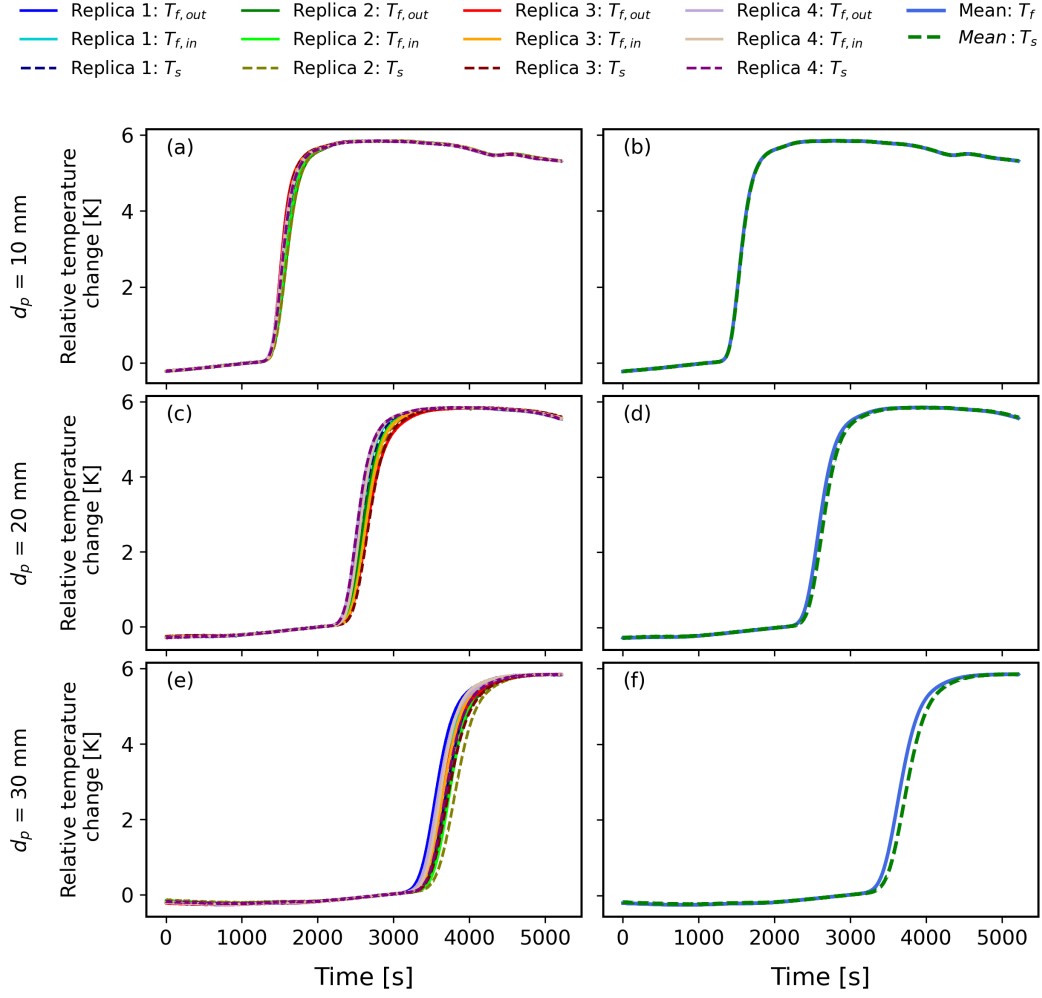

**Figure 3.** Thermal breakthrough curves (BTCs) derived from the processed data of temperature measurements (see Section 2.4) with a Darcy flux of 23 m d$^{-1}$, which shows the variation between measurements of LTNE probe replicas at the same depth. Here, $T_f$ is presented as solid lines, while $T_s$ is presented as dashed lines. $T_{f,in}$ and $T_{f,out}$ are outer $T_f$ measurements and inner $T_f$ measurement in a LTNE probe, respectively, indicating relative location of the fluid temperature measurements within the experimental column. (a and b) Corrected temperature measurements from all sensors and the averaged values of the corrected temperature for 10 mm grain at the depths of 45 cm are presented. (c and d) For 20 mm grain, corrected temperature measurements at the depths of 85 cm with deviation between all sensors and their averaged values including delay of thermal arrival in solid phase are illustrated. (e and f) For 30 mm grain as the largest tested grain, corrected temperature measurements with deviation among sensors of each phase and the averaged temperature with more pronounced deviation between fluid and solid are presented in comparison to (a-d).

pairs of solid and fluid measurements across all tested flow velocities, as depicted in Fig. 5. This negative $\Delta T(t)$ arises from the



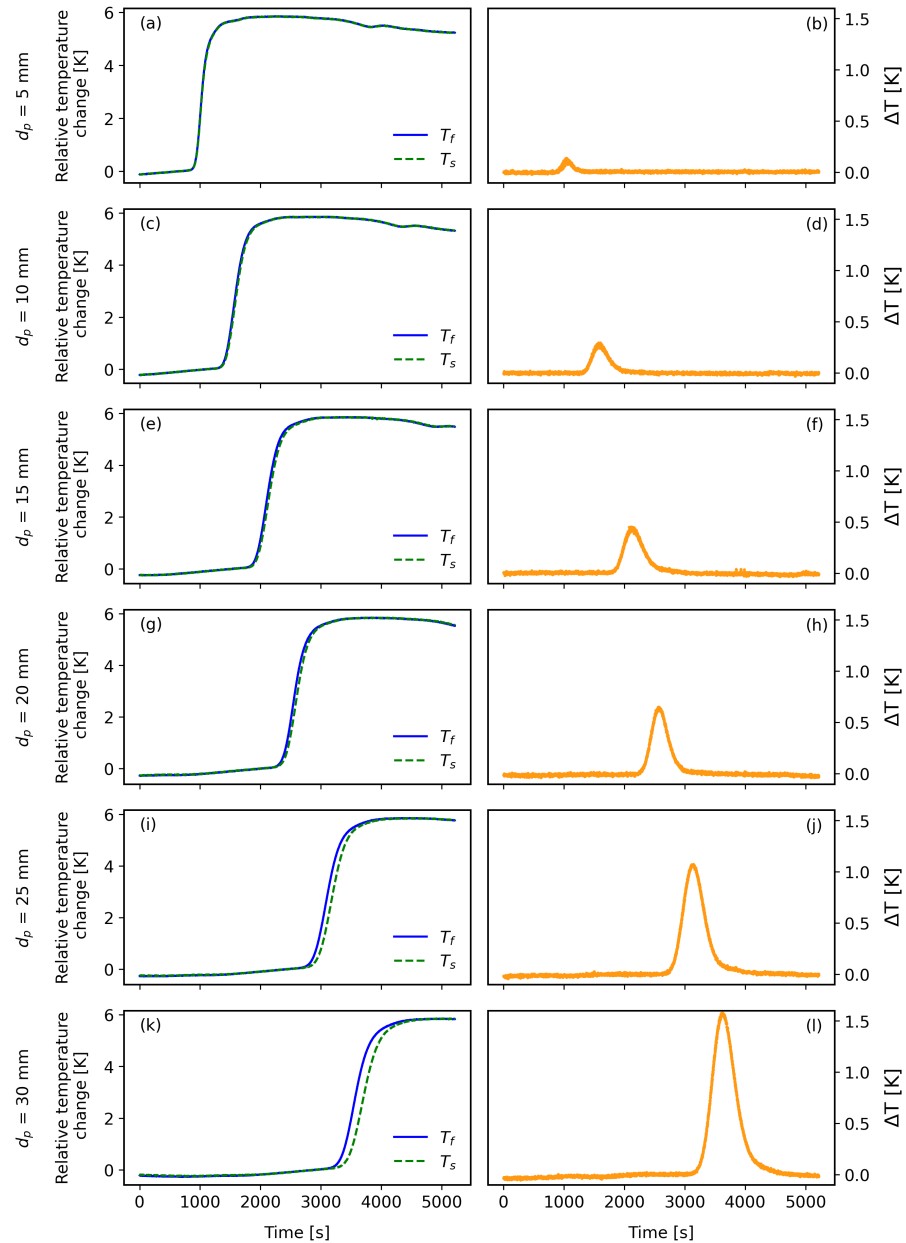

**Figure 4.** Thermal breakthrough curves (BTCs) of solid and fluid phase and $\Delta T(t)$ derived from experimental data with the maximum $\Delta T(t)$ among pairs of fluid and solid measurements in LTNE probe replica with a Darcy flux of 22.8 m d$^{-1}$. (a, c, e, g, i and k) Thermal BTCs of fluid and solid phases for each grain size. They display that the deviation between BTCs of $T_f$ and $T_s$ becomes larger with increasing grain size. (b, d, f, h, j and l) $\Delta T(t)$ for each grain size. They present an increase of $\Delta T(t)$ peaks with increasing grain sizes.



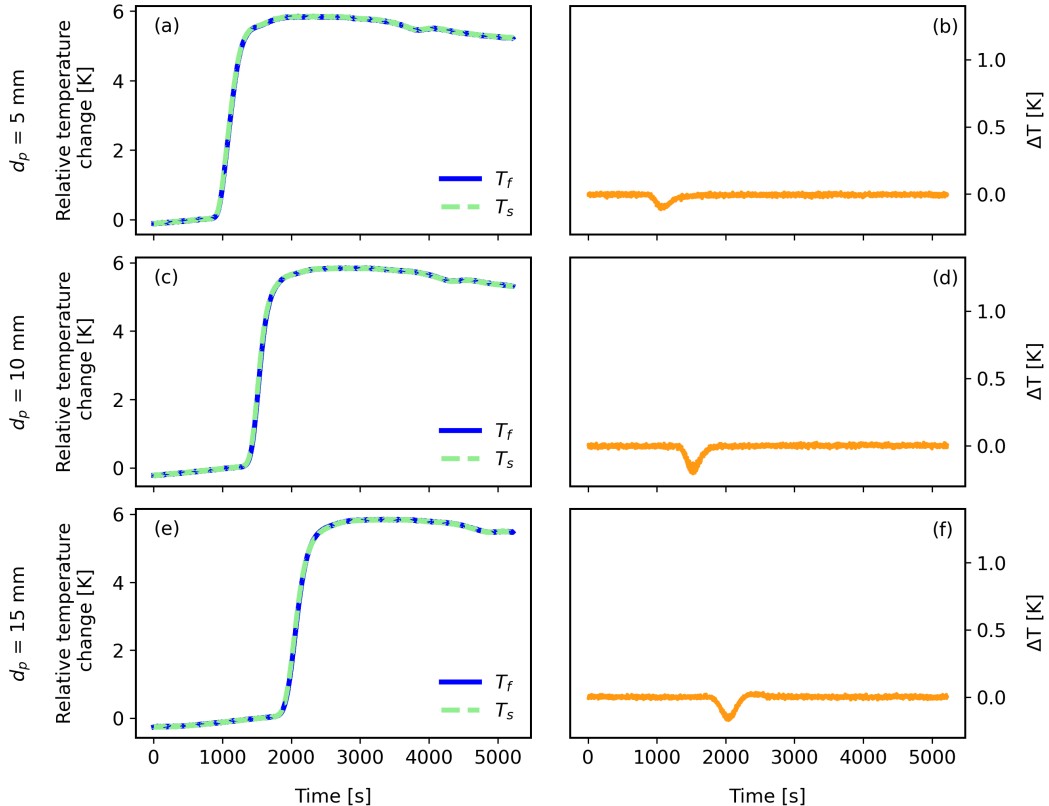

**Figure 5.** Experimental data showing examples of slightly faster thermal arrival of solid phase compared to the one of fluid phase as well as the corresponding 'inverse peaks' (i.e., negative $\Delta T(t)$) from the temperature difference between two phases for grain sizes between 5 and 15 mm at a Darcy flux of 23 m d$^{-1}$. This indicates the evidence of non-uniform flow effects in experiments. (a and b) Thermal breakthrough curves (BTCs) of fluid and solid phases and the corresponding inverse pulse for 5 mm grain. (c and d) Thermal BTCs of fluid and solid phases and the corresponding inverse pulse for 10 mm grain. (e and f) Thermal BTCs of fluid and solid phases and the corresponding inverse pulse for 15 mm grain.

solid phase exhibiting an earlier thermal response compared to the fluid phase, suggesting potential influences of a non-uniform
flow field resulting in different arrival times of the thermal front on both sides of the grain.

In Fig. 6, the LTNE effect is displayed for each of the six sphere sizes across all flow velocities. These $\Delta T(t)$ curves represent examples of pairs of fluid and solid measurements showcasing the highest maximum $\Delta T(t)$. In general, the LTNE effect intensifies with larger sphere sizes. Moreover, increasing velocities exhibit a consistent trend across all spheres, characterised by a heightened peak with earlier arrival time and a narrower spread of $\Delta T(t)$. This observation offers compelling evidence of
LTNE, facilitating the exploration of its relationship with grain size and flow velocity.

Figure 7 shows the quantitative evaluation of LTNE effects derived from the experimental data in relation to the grain size and flow velocity, based on the proposed classification approach in previous studies (Amiri and Vafai, 1994; Wang and Fox,



**Figure 6.** Summary of $\Delta T(t)$ curves from experimental data with all tested grain sizes from 5 to 30 mm diameter and Darcy velocities from 3 to 23 m d$^{-1}$. $\Delta T(t)$ curves are presented for each grain size with all tested Darcy velocities to compare the results with $\Delta T(t)$ curves of different grain sizes.

2023). The magnitude of LTNE effects can be determined by comparing the maximum normalized temperature differences.





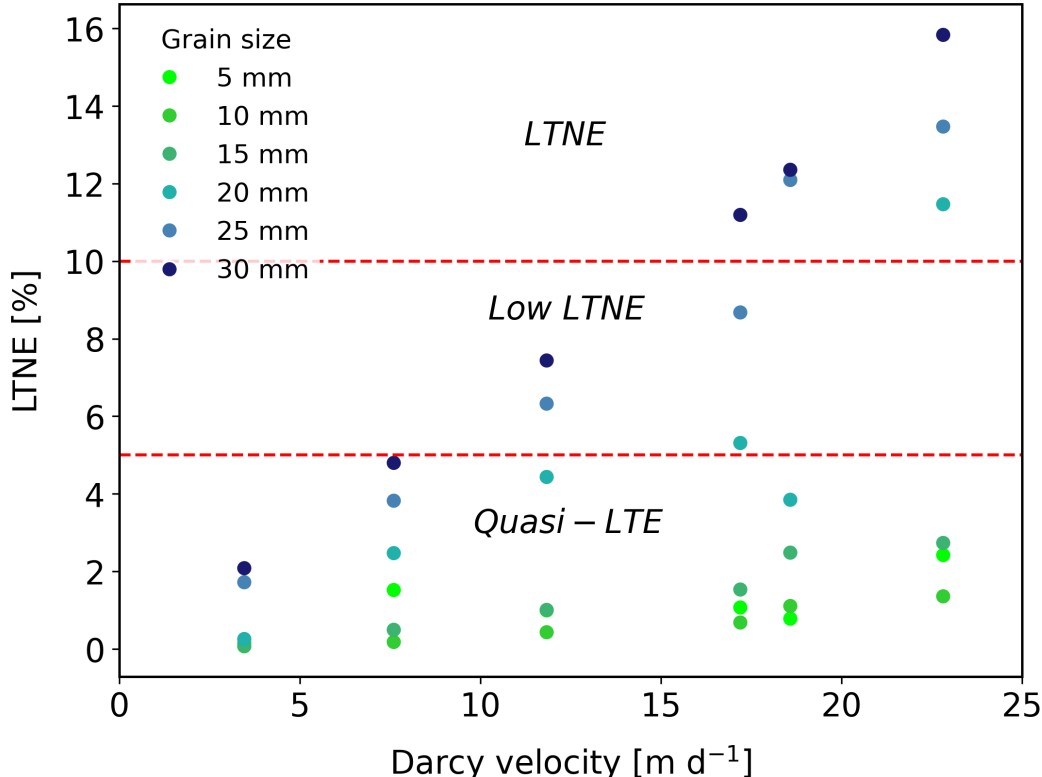

**Figure 7.** Quantitative evaluation of LTNE effects demonstrating the influence of grain sizes and flow velocities based on three categories: Quasi-LTE, < 5 %; Low LTNE, 5 - 10 %; LTNE > 10 %. The red dashed lines indicate LTNE lower limit for low LTNE (5 %) and LTNE (10 %). Experimental data for grain sizes ≥ 20 mm and Darcy velocity ≥ 12 m d$^{-1}$ revealed LTNE above 5 %.

This can be expressed as follows (Wang and Fox, 2023):

$$LTNE[\%] = 100 \times \frac{max \mid \Delta T(t) \mid}{T_1 - T_0} \tag{19}$$

This quantified LTNE is classified by three categories: Quasi-LTE, < 5 %; Low LTNE, 5 - 10 %; LTNE > 10 % (Fig. 7). This allows to compare LTNE effects from experiments where different boundary temperatures were applied. The results demonstrate that the LTNE effects become significant when flow velocity is > 12 m d$^{-1}$ for larger grain sizes > 20 mm.

### 3.4 Measured and modelled temperature breakthrough curves

The LTE analytical model exhibits limitations in predicting fluid temperature. This is particularly evident with larger grain sizes (≥ 20 mm) and faster flow velocities (≥ 12 m d$^{-1}$). Besides, it successfully models BTCs of measured fluid phase temperature for grain sizes of 5 mm and 10 mm, except for the tails of the BTCs. Notably, slower flow velocities (17 m d$^{-1}$) result in better fitting of modelled BTCs to experimental BTCs, as shown in Fig. 8a and b. However, discrepancies between





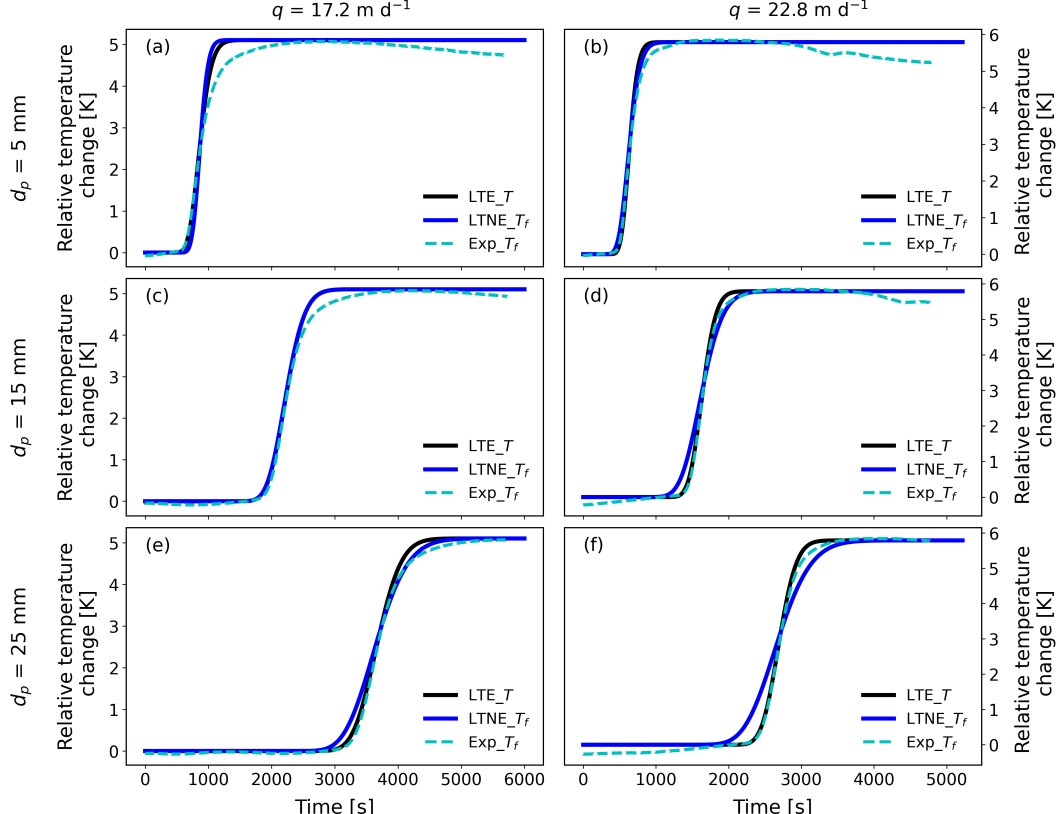

**Figure 8.** Comparison of LTE model, fluid phase results from the LTNE model, and experimental data across all grain sizes with Darcy velocities of $17.2 \, \mathrm{m \, d^{-1}}$ and $22.8 \, \mathrm{m \, d^{-1}}$.

measured fluid temperature and model predictions become more pronounced for the 15 mm grain size, especially at faster flow
velocities ($23 \, \mathrm{m \, d^{-1}}$), as illustrated in Fig. 8c and d. For grain sizes ranging between 20 mm and 30 mm, the LTE model can only predict the beginning of the fluid phase BTCs across all tested flow velocities.

The LTNE model, on the other hand, offers improved predictions for the tails of BTCs from experiments due to the larger spread of LTNE BTCs compared to LTE BTCs. While the LTNE numerical solution aligns well with the tails of BTCs from experiments at a flow velocity of $23 \, \mathrm{m \, d^{-1}}$, it displays an early rise at the beginning of the curves and relatively better fitting
at the end of the curves for slower flow velocities, as depicted in Fig. 8e and f.

The LTNE model demonstrated effective fitting to experimental BTCs and their corresponding LTNE effects for small grain sizes ranging from 5 to 15 mm, as depicted in Fig. 9. However, for grain sizes between 20 and 30 mm, the modeled $\Delta T(t)$ exhibited broader curves compared to experimental results. In the figure, LTNE model outcomes with $h_{sf}$ estimated by Eq. (12) - (15) (shown as green dash-dot lines) exhibited relatively good agreement with $\Delta T(t)$ curves from experiments for a grain





size of 5 mm, regardless of flow velocities (Fig. 9a-b). However, for grain sizes of 10 mm and 15 mm, the model overestimated $\Delta T(t)$ for all tested flow velocities, while it underestimated $\Delta T(t)$ for grain sizes ranging from 20 to 30 mm.

Nevertheless, the LTNE model successfully predicted the maximum $\Delta T(t)$ when the heat transfer coefficient was varied as a fitting parameter for all tested grain sizes and flow velocities, as depicted by the red lines in Fig. 9. However, for grain sizes between 20 mm and 30 mm, the LTNE model with fitted $h_{sf}$ struggled to match the BTCs and the spread of corresponding

$\Delta T(t)$ curves from experiments (Fig. 9h, j, l).

## 4 Discussion

### 4.1 Experiments reveal local thermal non-equilibrium heat transport

Our work utilized four separate LTNE probes at each distance along the flow path to capture the spatial variability of heat transport processes, thereby enhancing the interpretation of the experimental findings. By conducting separate temperature

measurements for both fluid and solid phases, we were able to discern the transient temperature disparities between these phases as $\Delta T(t)$. This demonstrated the occurrence of LTNE across various grain sizes (from 5 to 30 mm) and flow velocities (from 3 to 23 m d$^{-1}$) in a range between 0.018 K and 1.577 K, which is beyond the temperature sensor accuracy range of $\pm$ 0.01 K. While our observations are made for novel conditions, they align with the definition of LTNE by Kaviany (1995), that is characterized by considerable temperature differences between fluid and solid phases compared to the fluid temperature

difference over the system during advective heat transport in porous media.

The LTNE effects observed in our experiments confirm limited observations from previous experiments of heat transport in porous media with water flow. For example, by measuring fluid and solid temperatures separately, Levec and Carbonell (1985b) showed a delayed thermal pulse arrival in the solid phase for urea formaldehyde spheres ($\rho_s c_s = 0.002$ MJ m$^{-3}$K$^{-1}$, $\lambda_s = 1$ W m$^{-1}$K$^{-1}$), with a size of 5.5 mm. However, their work did not include an analysis of the temperature difference between the

two phases. With a similar two-phase temperature measurement approach, Bandai et al. (2023) demonstrated $\Delta T(t)$ derived from the temperature difference between two phases for 5 mm glass spheres. In the study by Bandai et al. (2017), they revealed the influence of particle size on thermal dispersion by heat transport experiments with small glass spheres (0.4 mm, 1 mm and 5 mm). While LTNE effects were determined without solid temperature measurement by estimating the effective thermal retardation factor when comparing solute and heat tracer experiments (Gossler et al., 2019; Baek et al., 2022), this approach

does not allow transient assessment and is therefore limited to qualitative determination of LTNE. Our study confirms LTNE effects under groundwater flow conditions and provides the ability to quantitatively determine transient LTNE effects as $\Delta T(t)$ in a relation to the grain sizes and flow velocities.

$\Delta T(t)$ was analyzed from all fluid and solid measurement pairs by subtracting the solid from the fluid temperatures. Due to the delayed thermal arrival of the thermal signal in the solid phases, $\Delta T(t)$ is expected to be positive always. However, negative

$\Delta T(t)$ resulting in an significant inverse pulse with the minimum between -0.31 and -0.04 were observed at some measurement locations for small grain sizes between 5 mm and 15 mm (Fig. 5). This phenomenon can be attributed to non-uniform flow, an experimental observation that was previously reported when multiple temperature sensors were used at the same discrete

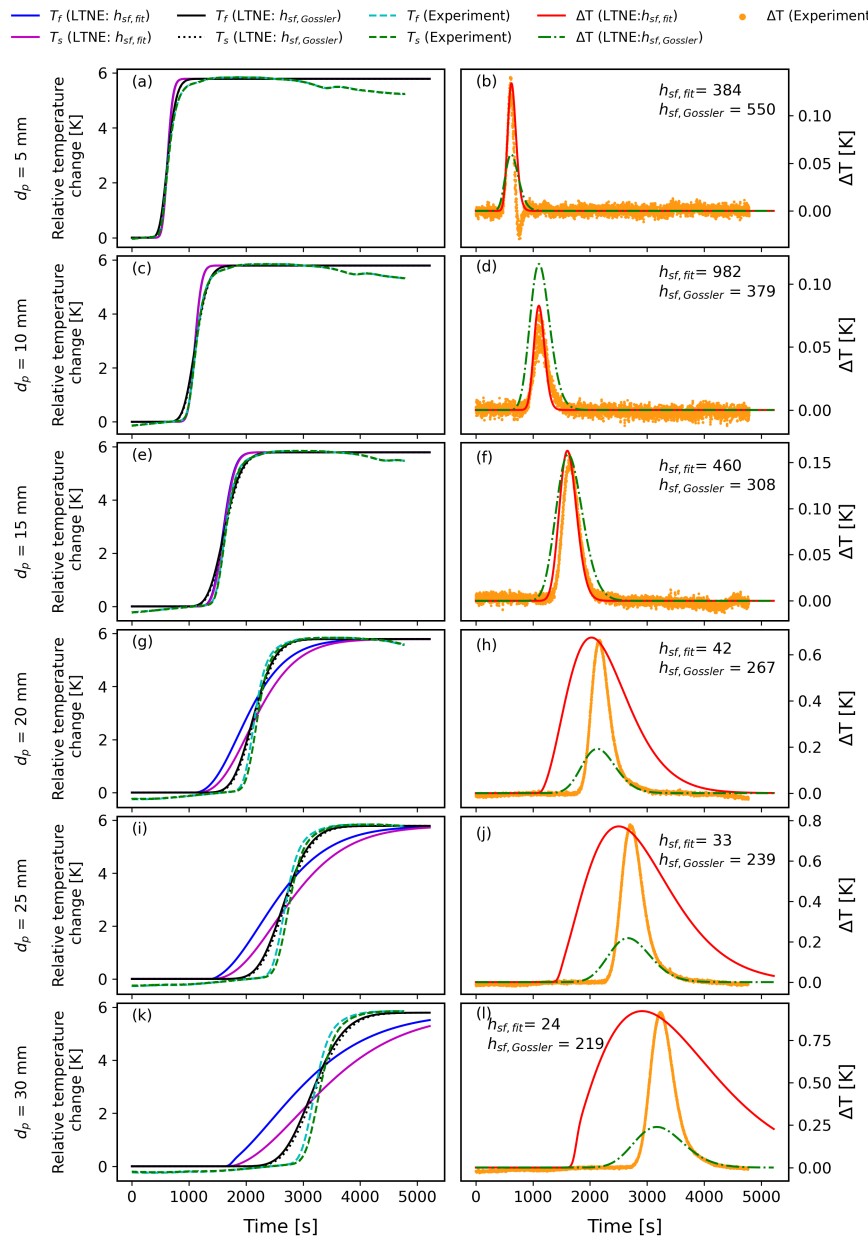

**Figure 9.** Comparison of experimental data with LTNE model outcomes using varied $h_{sf}$ for all tested grain sizes at the highest flow velocity (23 m d$^{-1}$). LTNE model was simulated with the estimated $h_{sf}$ by the correlation of Gossler et al. (2020), $h_{sf,Gossler}$, and by fitting to the experimental data, $h_{sf,fit}$. (a, c, e, g, i and k) Thermal breakthrough curves (BTCs) of fluid and solid phases for 6 different grain sizes derived from experiments and two LTNE model outcomes with $h_{sf,Gossler}$ and $h_{sf,fit}$. (b, d, f, h, j and l) $\Delta T(t)$ for 6 different grain size from LTNE model with $h_{sf,Gossler}$ and $h_{sf,fit}$. The estimated $h_{sf}$ value for each model is presented for each grain size.





locations along the flow path (Rau et al., 2012b). Non-uniform flow causes the thermal front to propagate non-uniformly in the transversal, i.e., perpendicular to the flow direction. This means that local thermal velocities at the thermal front are different.

In our case, non-uniform flow causes the thermal front to arrive at different times on both sides of a sphere, leading to the solid response being faster than the fluid response on the side with slower velocity. The result is a negative $\Delta T(t)$ hereafter referred to as an 'inverse pulse'. The occurrence of this phenomenon for smaller grain sizes suggests that this flow non-uniformity may either occur at small scales and/or be spread out through transverse dispersion over travel distance disallowing detection. Notably, inverse pulses of $\Delta T(t)$ were not observed for grain sizes of 20-30 mm, suggesting that non-uniform flow may have

a stronger impact on results with smaller grain sizes, owing to the smaller representative elementary volume (REV).

Previous studies that conducted experiments with separate temperature measurements for the two phases demonstrated LTNE effects were limited to grain sizes $\leq 5.5$ mm (Levec and Carbonell, 1985b; Bandai et al., 2023). In the study of Bandai et al. (2023), temperatures for fluid and solid phases were separately measured and the maximum temperature difference between fluid and solid phases for 4.94 mm glass spheres with thermal conductivity of 0.76 W m$^{-1}$K$^{-1}$ was up to 0.04 K at a Darcy

velocity of 29 m d$^{-1}$. In comparison, our study showed a higher maximum temperature difference of 0.14 K between the two phases for 5 mm spheres with a lower Darcy velocity of 23 m d$^{-1}$. This discrepancy could be due to non-uniform flow degrading the magnitude of $\Delta T(t)$ compared to uniform flow. The same mechanism could also cause LTNE effects with stronger magnitude that are caused by local differences in the thermal velocity surrounding the sphere due to non-uniform flow effects. Having four replicas for each sphere size provides the advantage of capturing the variability and allowing more robust

assessment of LTNE.

### 4.2 Local thermal non-equilibrium increases with grain size and velocity

Using $\Delta T(t)$ as a measure for transient LTNE allows detailed insight into the heat transport processes. Our results clearly show that LTNE effects increase in magnitude with grain size ranging from 5 to 30 mm and Darcy velocities ranging from 3 to 23m d$^{-1}$. The wider $\Delta T(t)$ peaks observed at slower flow velocities indicate that it takes longer to achieve thermal

equilibrium between the two phases with lower flow velocities. Furthermore, for Darcy velocities ranging from 3 to 23 m d$^{-1}$, the magnitude of $\Delta T(t)$ grows up to about 10 times of a 5 mm grain size with increasing grain size, showing the stronger LTNE effects for larger grain sizes for all tested flow velocities.

While inverse pulse of $\Delta T(t)$ were observed for 5–15 mm grain sizes across all tested flow velocities, the maximum $\Delta T(t)$ in the experiments tended to be higher than the minimum $\Delta T(t)$ in inverse pulse. Notably, the magnitude of LTNE effects for

the smallest grain size of 5 mm remains smaller than 0.2 K for all tested flow velocities. This illustrates that the influence of flow velocities on LTNE for the smallest grain size of 5 mm was not clearly evident in our study, which aligns with recent theoretical investigations hypothesizing that LTNE effects should not occur for grain sizes smaller than 7 mm (i.e. for sand and fine gravels) (Gossler et al., 2020).

We note that Baek et al. (2022) identified LTNE effects for a grain size as small as 0.76 mm but with fast Darcy velocities

that exceed 20 m d$^{-1}$. However, they did not directly measure solid and fluid temperatures, but instead established LTNE by comparing solute with heat transport. In our study, no significant increase in LTNE effects was observed for a 5 mm grain size.




This discrepancy could be attributed to heterogeneity of porous media in different grain sizes and shapes as reported by Baek et al. (2022).

### 4.3 Simplified heat transport models insufficiently describe local thermal non-equilibrium

We replicated our experimental observations using LTE and LTNE models which led to mixed results. While the LTE model can be adjusted to fit near the beginning of breakthrough curves (BTCs) by varying thermal velocity and dispersion coefficient, it fails to adequately model the entire BTC, including both the beginning and the tail. Bandai et al. (2023) also conducted heat transport experiments measuring fluid and solid phases separately and observed that the tail of BTCs from the fluid phase were more spread out compared to the LTE model, likely due to a non-ideal step heat input. While our temperature measurements
from the top of the porous media exhibited steep BTCs in Fig. 2, they differed from the ideal step input (Heaviside step function) required to comply with the model's boundary conditions. This may lead to a misrepresentation of heat transport parameters from misfitting.

The LTNE model was utilized to predict the magnitude of LTNE effects, determined by $\Delta T(t)$. The maximum $\Delta T(t)$ can be adjusted by varying the heat transfer coefficient as a fitting parameter in the model. The estimation of the heat transfer
coefficient by correlation of Gossler et al. (2020) was unable to model the maximum $\Delta T(t)$. This could be caused by the empirical relationship between $Nusselt$ number $Nu$ and $Reynolds$ number $Re$ to derive the heat transfer coefficient. Consequently, the empirical $Nu$ could lead to over- and underestimation of $\Delta T(t)$ by changing the spread of modelled BTCs. Our modeling results show that the 1D LTNE model closely describes the temperature difference between the fluid and solid phases for grain sizes of 5 mm and 10 mm (Fig. 9). However, for a grain size of 15 mm, deviations from experimental breakthrough
curves (BTCs) become larger. For larger grain sizes ranging from 20 to 30 mm, the deviations become significant, and the LTNE model is unable to accurately predict $\Delta T(t)$. When optimizing the fitting of the maximum $\Delta T(t)$ by adjusting the heat transfer coefficient, the BTCs of the model deviate further from the experimental BTCs, deteriorating the fitting (Fig. 9g, i, k). This limitation may be attributed to the constraints of the 1D model not capturing the multi-dimensional processes as caused by non-uniform flow evidenced earlier.

Overall, non-uniform propagation of the thermal front caused by non-uniform flow leads to temperature gradients in the transverse direction and influences the nature such as the magnitude of $\Delta T(t)$. Unfortunately, such processes cannot be captured by a 1D LTNE model, as this is limited to describe the heat transport in the flow direction only. This does not accurately represent the experimental setup and exact temperature measurement points for fluid and solid phases. Consequently, to determine transport parameters such as the heat transfer coefficient ($h_{sf}$) from our experimental datasets, more sophisticated LTNE
models are required. This goes beyond the scope of our study and should be done in future work.

### 4.4 Implications for modelling heat transport in porous aquifers

Our experimental work confirms the presence of LTNE effects, prompting inquiry into their relevance to groundwater flow in aquifers. The glass spheres we employed possess a thermal conductivity of 1 W m⁻¹K⁻¹ and a volumetric heat capacity of 1.9 MJ m⁻³K⁻¹. While these values may deviate from typical thermal parameters of groundwater systems, they fall within the





**Table 2.** Comparison between thermal properties of natural material (rock) from literature (Clauser, 2021a, b) and experimental material (glass).

| Parameter | Glass | Rock | | | Unit | Source |
|---|---|---|---|---|---|---|
| | Measurement | min | average | max | | |
| Thermal conductivity of solid $\lambda_s$ | 1.0 | 0.4 | 4.1 | 7.9 | W m$^{-1}$K$^{-1}$ | Menberg et al. (2013) |
| Volumetric heat capacity of solid $\rho_s c_s$ | 1.9 | 1.3 | 2.3 | 3.4 | MJ m$^{-3}$K$^{-1}$ | Clauser (2021a) |

reported range for natural sediments (Table 2). For instance, thermal conductivity values range from 1 to 7.9 W m$^{-1}$K$^{-1}$ for sedimentary rocks and quartz mineral, respectively (Clauser, 2021b; Menberg et al., 2013), and volumetric heat capacity values range from 2.3 to 3.6 MJ m$^{-3}$K$^{-1}$ for impervious rocks and inorganic minerals, respectively (Banks, 2015; Clauser, 2021a). In the study by Bandai et al. (2023), they utilized an LTNE model to compute $\Delta T(t)$ across various thermal conductivity values (ranging from 0.23 to 2.3 W m$^{-1}$K$^{-1}$) and volumetric heat capacity of the solid phase (ranging from 1.0 to 4.18 MJ m$^{-3}$K$^{-1}$).

Their findings indicated that thermal conductivity does not significantly influence LTNE; i.e., the magnitude of $\Delta T(t)$ remains relatively stable. However, an increase in the volumetric heat capacity of the solid phase leads to heightened LTNE. This phenomenon occurs because the solid phase requires more energy to achieve a similar temperature rise. On the contrary, Gossler et al. (2020) theoretically demonstrated that the volumetric heat capacity of the solid phase exerts minimal influence on LTNE effects within an LTNE numerical model by means of parameter sensitivity analysis. To decipher the implications for

real-world systems like porous aquifers, addressing this disparity demands the creation of sophisticated models that accurately represent the experimental heat transport processes.

     To try and explain our experimental observations and gain insight into thermal properties and processes we used standard models accepted in the literature. However, our results indicate that the LTE model cannot distinguish between the fluid and solid and is therefore limited to simplified heat transport scenarios without considering temperature differences between phases.

Additionally, our simple 1D LTNE model failed to adequately represent the measured $\Delta T(t)$. The analysis revealed three main factors that were identified as limiting: (1) our measured BTCs did likely not comply with the ideal boundary condition (Heaviside step function) assumed by standard analytical solutions, (2) the occurrence of non-uniform flow caused inverse pulses and may therefore also contribute to variations in $\Delta T(t)$ that cannot be captured by simple models, and (3) LTNE heat transport appears to be a multi-dimensional process with geometrical effects. This clearly highlights the limitations of simplified heat

transport models to estimate thermal parameters and capture advanced heat transport processes from experiments. We suggest that future studies focus on developing advanced numerical models capable of incorporating a greater level of detail. These models should be adopted at analyzing experimental data and providing deeper insights into the intricacies of heat transport processes.





## 5 Conclusions

We conducted systematic laboratory experiments on heat transport by subjecting water flow to temperature step inputs at Darcy velocities ranging from 3 to 23 m d$^{-1}$ through porous media composed of idealized spherical grains with diameters between 5 and 30 mm. Temperature breakthrough curves (BTCs) were separately measured in the fluid and solid phases. Our results unequivocally demonstrate transient local thermal non-equilibrium (LTNE) heat transport effects, characterized by a temporary temperature discrepancy $\Delta T(t)$ between the two phases over time. This discrepancy indicates that the solid phase exhibits a

time lag compared to the fluid phase in response to passing thermal transience. Importantly, we observed that the LTNE effect becomes more pronounced with increasing grain size (5 - 30 mm) and Darcy velocity (3 - 23 m d$^{-1}$), aligning with theoretical predictions yet previously unverified. Furthermore, negative temperature differentials between the solid and fluid phases for smaller grains (5 - 15 mm) were attributed to non-uniform flow inducing transverse temperature gradients.

To reconcile experimental observations and estimate heat transport parameters, we employed both an analytical solution

assuming local thermal equilibrium (LTE) heat transport and a numerical solution to the transient local thermal non-equilibrium (LTNE) differential equations, both of which are state-of-the-art and conducted in one-dimensional space. The LTE model exhibits relatively good agreement with the breakthrough curves (BTCs) observed in the fluid phase for small grain sizes ranging from 5 to 15 mm. However, for larger grain sizes ($\geq$ 20 mm), the LTE model fails to adequately describe heat transport, primarily due to significant LTNE effects. Additionally, discrepancies between the models and experimental data in

the tail of BTCs for large grains suggest that the experimental conditions may not align with the boundary conditions assumed in the solution. Analysis of the experimental data using the LTNE model yields successful results only for small grain sizes within the range of 5 to 15 mm, while the model struggles to accurately capture transport behavior for larger grain sizes ($\geq$ 20 mm) such as coarse gravels.

The experimental findings from this study provide experimental evidence for grain size and velocity dependent transient lo-

cal thermal non-equilibrium (LTNE) effects that was postulated theoretically. However, a comprehensive comparison between experimental data and models reveals only partial success. Several factors contribute to this discrepancy: (1) Non-ideal boundary conditions, deviating from the assumed step-like conditions in standard analytical solutions, are present in the experiments. (2) Non-uniform flow induces inverse temperature gradients, altering $\Delta T(t)$ and complicating the interpretation of properties from BTCs. (3) State-of-the-art one-dimensional models lack the capacity to fully capture the multi-dimensional nature of

LTNE heat transport processes.

Future research endeavors should prioritise the development of sophisticated two-phase numerical models capable of analyzing the experimental dataset comprehensively, enabling the derivation of advanced heat transport processes and properties.

*Data availability.* The experimental dataset and analysis scripts will be made available after successful peer review.



*Author contributions.*  HL designed the experiment with guidance from MG. HL built the experimental setup with some assistance from MG
and conducted all experiments. HL analysed the experimental datasets and wrote the manuscript draft. KZ, PBa and PBl provided feedback
on the draft manuscript. GCR supervised HL throughout the entire process. GCR and PBa developed the working hypotheses, conceptualized
the research plan and received the funding that supported this work.

*Competing interests.*  The authors declare no competing interest.

*Acknowledgements.*  This project has received funding from the German Research Council (DFG) grant agreement number 468464290.



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
