# Peer review of "Laboratory heat transport experiments reveal grain size and flow velocity dependent local thermal non-equilibrium effects"

_EGUsphere, 2024_

## Author Comment (AC1)

**Response to Community Comments 1 (Responses in blue)**

*General comments*

*Good theoretical research that can be improved addressing the specific comments below.*

Thank you for your effort in reading our work and providing comments that point out descriptions to be improved. We will consider your comments carefully and provide further clarifications on the revised manuscript. Please find detailed replies below.

*Specific comments*

*Line 23 "Accurately describing heat transport in porous media has long been a focus in both engineering and science". Insert recent review papers on heat transport in geological media since the sentence is not backed-up by references.*

*- Review of discrete fracture network characterization for geothermal energy extraction. Frontiers in Earth Science, 11, p.1328397*

*- Review of geothermal energy resources, development, and applications in China: Current status and prospects. Energy, 93, pp.466-483*

Thank you for suggesting related references. We will include them in our work as suggested.

*Line 27. Specify low enthalpy geothermal systems?*

The geothermal systems of our interest for the research are the systems, which utilize groundwater, such as groundwater heat pump (GWHP) systems and aquifer thermal energy storage (ATES) systems. We will clarify this in our revised manuscript.

*Line 78. The aim is clear, but please specify the 3 to 4 objectives of your research by using numbers (e.g., i, ii and iii).*

We will revise our sentences to clarify the objectives of this research.

*Line 153. "Porous media" provide more detail on the material that you used to create the porous material that approximate the geological media in your analogue.*

In this study, porous media was represented by packed glass beads in a vertical column, as it is described in Fig. 1. We will refine our formulation.

*Lines 308-424. Provide more detail on the validity of your analogue experiment that is at small scale. Much smaller than the aquifer. This point can be addressed here or in the introduction.*

Understanding LTNE effects starts from the grain scale. By detecting LTNE effects which are caused by temperature difference between fluid in pores and solid as grains, thermal disequilibrium caused by LTNE can be determined. As this phenomenon is ambiguous and occurs together with hydromechanical dispersion effects at larger scale, the experiments at the small scale provides the information on heat transfer between fluid and solid phases. The results in this study can provide insights for the significance of LTNE effects at larger scales. We will clarify this in our revised manuscript.

*Lines 308-424. Provide more detail on the validity of your analogue research taking into account that porous aquifers (typically siliciclastic) are very heterogeneous. This point can be done here or in the introduction.*

The experiments in this study aimed to evaluate LTNE effects at the grain scale under varied flow velocities. Although the relation between LTNE effects and each grain size was revealed in this study, this finding would contribute as a fundamental knowledge to understanding of LTNE effects in heterogeneous porous media for future research. We will clarify this in our revised manuscript.

*Figures*

*Figure 1a. Insert the spatial scale.*

The spatial scale will be included in the revised manuscript.

*Figure 1d. Insert the spatial scale.*

The spatial scale will be included in the revised manuscript.

*Figures 3, 5 and 8. Room to make the figures larger.*

The figures will be enlarged in our revised manuscript.

---

## Author Comment (AC2)

**Response to Reviewer Comments 1 (Responses in blue)**

*To investigate the presence of local thermal non-equilibrium (LTNE) effects during heat flow in porous media, in the present work, laboratory experiments systematically investigated heat transport by exposing water flow to temperature step inputs. The experiments were conducted at Darcy velocities ranging from 3 to 23 m/d through porous media comprising idealized spherical grains with diameters between 5 and 30 mm. This work is new and interesting, I think this paper can do well in the future. I have a few specific comments that I will list below in order to improve the current manuscript.*

Thank you for pointing out the key messages and seeing the positive prospective in our manuscript. We will address all of your questions to improve our manuscript. Please find a detailed reply to each of your questions below.

*Lines 68-74: Yes, there are very few experimental studies on LTNE in porous media, and in addition to the two studies mentioned by the authors, laboratory experimental studies have been conducted in the study of Shi et al. (2024) [DOI:10.1029/2024WR037382], and it is suggested that the authors refer to their experimental studies section for an introduction as well.*

We agree that the newly published Shi et al. (2024) contributed to the laboratory experiment results and will incorporate their findings in our revised manuscript.

*Line 163: "thermal conductivity" -> "thermal conductivities"*

The word will be corrected in the revised manuscript.

*Line 165: "specific heat capacity" -> "specific heat capacities"*

The word will be corrected in the revised manuscript.

*Line 173: "dispersion coefficient" should be changed to "thermal dispersion coefficient" as "dispersion coefficient" is a part of "thermal dispersion coefficient".*

This term will be corrected in the revised manuscript.

*Eq. (14): Heinze (2024) [DOI: 10.1016/j.earscirev.2024.104730] provides a detailed overview of this relationship, and I would suggest that the authors could make appropriate references to the work of Heinze (2024) here. Just a suggestion:)*

We agree and will incorporate the finding by Heinze (2024) especially the recent overview of Nusselt number in terms of heat transfer in porous aquifer.

*Line 321: "In the study by Bandai et al. (2017), they revealed…"-> "Bandai et al. (2017) revealed…".*

We will revise this as suggested.

*Lines 412-413: Please express this sentence in two sentences.*

We will revise this as suggested.

---

## Author Comment (AC3)

**Response to Reviewer Comments 2 (Responses in blue)**

*The manuscript presents novel heat transport/heat transfer experiments that address a long-standing research gap on the relevance of LTNE effects in aquifer systems. The manuscript is a highly valuable contribution to the research topic and provides great insights into the microscopic temperature distribution in large grain size sediments, building on previous experiments by some of the authors.*

Thank you for your positive comments pointing out the value of our research. All of your comments will be considered carefully to improve our manuscript. The detailed replies are provided below each of your question.

*Abstract: It would be great if the abstract could include some more quantitative values and clarifications. Examples:*

*- "some temperature differences were negative" -> the reader doesn't know what positive/negative temperature differences mean at this point.*

We will correct the sentence by clarifying the results of inverse pulse ΔT, which are likely caused by delayed thermal arrival in fluid phase by the local advective flow.

*- "relatively good agreement", "became too significant" -> can these statements be quantified?*

The quantitative evaluation of the model fit will be included in the revised manuscript by providing the values of root mean squared error (RMSE) for each model result.

*- I believe that the term "magnitude of LTNE" is not universally known, given the wide readership of HESS. Therefore, I propose to define the term if it is used in the abstract.*

Thank you for your suggestion. The term of "Magnitude of LTNE" was used to describe the peak of ΔT(t) curve. Considering your comment, the term will be improved to better reflect its meaning in the revised manuscript.

*Materials and methods*

*\* The authors state at the beginning that "specialised experimental instrumentation" was used, but it remains unclear in the rest of the text what this "specialised instrumentation" is and how the current procedure was "adapted" from Gossler et al. (2019). The lessons learned from Gossler et al. (2019) are nicely highlighted in the following paragraphs.*

In this study, only one refrigerated bath circulator was used to prepare warm water as a heat source. To establish initial temperature, water circulation with outlet water was performed in this study as it is stated in line 126-127, while Gossler et al. (2019) provided cold water from a refrigerated bath. In addition, LTNE probes with Pt100 type A sensors are newly introduced in this study. We will improve this in the revised manuscript.

*\* Why were all samples measured in one experimental setup and stacked on top of each other? Why not test all grain sizes separately? Did the authors reverse the order of the spheres to check for a possible influence?*

Previous research has explored the relationship between LTNE effects and hydrogeological properties such as flow velocities and grain sizes. This study aims to investigate how varying grain sizes influence heat transport under different flow conditions. To do this, we drew on deep insights from previous studies (Rau et al., 2012a; Gossler et al., 2019; Bandai et al., 2023), when designing our experiment to effectively measure the temperature difference between fluid and solid phases. We decided that our experimental setup is particularly suited for testing heat transport for the following reasons:

- Placing all six different grain sizes into one experiment allows experimentation with identical flow velocities. This is difficult to do in separate experiments.
- We arranged the spheres in the column based on the spatial gradient of the thermal front. As heat propagates downward through the column, the spatial derivative (or steepness) of the thermal front decreases. Consequently, the smallest grain size, expected to have the least LTNE effects (Gossler et al., 2019), was placed at the beginning of the column to respond to the steepest gradient. Larger grain sizes were then arranged progressively deeper in the column.
- Experimentation is labour and time intensive. Overall, this approach provided us with the ability to efficiently experiment with two degrees of freedom (grain size and velocity) in one experimental setup.

We hope this clarifies your questions and welcome any further discussion or specific questions for additional clarification

*\* When filling the layers with the small glass beads, was it necessary to keep the outer PT100 free from contact with the smaller beads or did the authors rely on a (rapid) LTE between the small glass beads and water?*

PT100 sensors for fluid phase are surrounded by small glass beads without additional structure to keep them away from the beads. Although it might cause contact between the sensors and the glass beads, we assume that fluid phase and solid phase of small glass beads (dp = 1 mm) establish an instant thermal equilibrium (LTE) resulting in the same temperature. We relied on a rapid thermal equilibrium between the glass beads and water. This is justified given the findings by Gossler et al. (2019) as well as our own results.

*Analysis/Results*

*\* What effect did the "correction" steps (pp. 225-229) have on the results? What was the difference/spread between the sensors? Should this be shown in Fig. 3? The curves in Fig. 3 are almost indistinguishable visually. In particular, which would be of interest, it is not possible to see whether the difference between Ts and Tf is consistent but shifted across the probes. Was there a systematic relationship between the spread and grain size or flow velocity? I would encourage the authors to try a different way of visualising Figure 3,*

*perhaps adjusting the range of the x-axis or visualising differences rather than absolute values.*

The data was corrected to make the temperature difference between fluid and solid phases comparable. This allowed to determine streamlined LTNE effects as ΔT curves with a peak. Fig. 3 is presented to show comparison between temperature measurements from all sensors at the same depth. Considering your comments, the relationship between the spread and grain size or flow velocities will be analysed, and the further explanation will be included in the revised manuscript. Also, we will consider an improved presentation for Fig. 3 as suggested.

*\* The wording in the text and figures does not seem to be consistent throughout the text. For example, Figure 2 refers to "calibrated temperature", which is not used in the text. Also, "inner" and "outer" Tf measurements are not explained in the text (Fig. 3) - could this be done in Fig. 1?*

We will remove these inconsistencies in our revised version of the manuscript.

*\* The relevance of the negative temperature pulses is difficult to assess. l. 267f states: "was observed in some pairs of (...) measurements over all (...) flow velocities". What does this mean? Some replicates showed negative temperature pulses while others showed positive pulses? Because this is associated with the smallest grain sizes: Could this be an experimental problem, as the Pt100 sensor is the same size for all grain sizes, so its placement within the glass beads is more sensitive in smaller beads? Also, l. 270: "both sides of the grain" suggests that there is a discrepancy between the Pt100 sensors to the left and right of the glass beads. This would be interesting to see rather than Fig 5a,c,e which shows pretty much two lines on top of each other. Does this grain size discrepancy only occur for the small grains or is it only of less significance for the larger glass beads? Is this what you want to show in Fig. 3?*

Thank you for your comments with detailed questions. Although we have fixed the sensor position using the PVC frame and placed at the specific depth inside column as shown in Fig. 1., the possibility of sensor misplacement cannot be eliminated. We will further analyse the data investigating the relationship between the sensor positions and inverse curve results in our revised manuscript. Additionally, the sensitivity of small grain sizes to the Pt100 will be investigated taking your comment into consideration.

*Discussion*

*\* Can the "agreement" between numerical models and experiments be quantified in some way to quantitatively demonstrate the "better fit"?*

As it is stated earlier, we will add the root mean squared error (RMSE) for each model to the revised manuscript.

*\* The authors measure the temperature in the middle of the glass beads and refer to it as "solid temperature". However, within the glass beads there will be a radial thermal gradient. The "mean solid temperature" across the glass beads might be higher than the temperature*

*measured observed by the authors. Hence, the heat transport within the larger grains takes longer time due to the larger distance from the outside of the glass bead to the temperature sensor. At the contact surface of the grain and the water, there still might be thermal equilibrium. While the subsequent theoretical considerations in terms of the influence of flow velocity go beyond the scope of this work, a short discussion on the scale effect could enrich the Discussion section.*

Temperature for solid phase was measured at the centre of glass spheres to represent solid temperature. The surface of glass sphere was technically difficult to measure due to the thickness of the sensor and the small contact area of each glass sphere. The challenges and inconsistency of the surface solid phase temperature measurement have been reported in the study of Bandai et al. (2023), which conducted heat transport experiments measuring solid phase temperature. We will add a short discussion of the implications to our discussion in the revised manuscript.

**References**

Bandai, T., Hamamoto, S., Rau, G. C., Komatsu, T., & Nishimura, T. (2023). Effects of thermal properties of porous media on local thermal (non-)equilibrium heat transport. Journal of Groundwater Hydrology, 65 (2), 125-139. doi: https://doi.org/10.5917/jagh.65.125

Gossler, M. A., Bayer, P., & Zosseder, K. (2019). Experimental investigation of thermal retardation and local thermal non-equilibrium effects on heat transport in highly permeable, porous aquifers. Journal of Hydrology, 578, 124097. doi: https://doi.org/10.1016/j.jhydrol.2019.124097

Rau, G. C., Andersen, M. S., & Acworth, R. I. (2012a). Experimental investigation of the thermal dispersivity term and its significance in the heat transport equation for flow in sediments. Water Resources Research, 48 (3). doi: https://doi.org/10.1029/2011WR011038

---

## Author Comment (AC4)

**Response to Reviewer Comments 3 (Responses in blue)**

*The paper "Laboratory heat transport experiments reveal grain size and flow velocity dependent local thermal non-equilibrium effects" conducted laboratory experiments to clarify thermal non-equilibrium between solid and liquid phases in saturated porous media under forced convection. This study provides valuable laboratory experimental data on the effects of grain size and flow rates on thermal non-equilibrium between solid and fluid phases of saturated porous media. The authors measured not only the fluid temperature but also the temperature of solid phase by specifically designed probes, which only a few studies have reported before.*

*Regardless of the value of the experimental data, I have a few concerns regarding data analysis and interpretation of the experimental data. I believe the manuscript could be improved by re-analyzing the same data without additional experiments. Therefore, I recommend a major revision for potential publication of this manuscript.*

Thank you for your detailed review which has brought up good questions. We will consider all of your comments carefully to improve our manuscript. Please find replies below each comment.

*Major and minor points are summarized below (numbers indicate the line numbers).*

*Major points*

*Choice of LTNE model*

*While the authors used the LTNE model (Eq. 10 and Eq. 11), this LTNE model may not be applicable to the experimental data. The LTNE model with spatially uniform parameters (e.g., h_sf and lambda_s, eff) assumes that the physical property of solid phase is uniform in the spatial domain. However, in the experimental setup, glass spherical particles with varying sizes were embedded in finer glass spheres of 1 mm diameter. Although some of the parameters can be justified to be spatially uniform even in this setting (e.g., thermal conductivity of solid and porosity), this setting violates the assumption of the LTNE model. For example, the heat transfer coefficient and surface area are functions of grain sizes. The LTNE model with spatially uniform parameters is applicable when the porous media is made up of uniform grain sizes (could be non-uniform up to the validity of representative elementary volume).*

*If the authors (or the editor and other reviewers) want to include the analysis with an LTNE model, I would recommend a LTNE model presented in Wakao and Kaguei, 1982, where energy equation for a solid spherical particle is coupled with energy equation for fluid phase.*

*In this way, the authors can simulate LTNE of a spherical particle embedded in saturated porous media.*

*Wakao, N. and Kaguei, S. (1982): Heat and mass transfer in packed peds. Gordon and Breach Science Publishers, Inc, 364p.*

Thank you for your suggestions regarding the LTNE model. We agree that the LTNE model in this study is limited to applications with uniform grain sizes. We will add the suggested model to our revised manuscript.

*Calibration of temperature data*

*Regarding Line 225-229, it is more natural to normalize the measured temperature by the temperature difference between the initial and final temperature, as in Eq. 19, not by the final temperature. Doing a proper temperature calibration might provide temperature data, that is more compatible with the LTE model:*

We will do this as suggested in our revised manuscript.

*- In Figure 2, what caused the increase in the calibrated temperature at deeper depths before the arrival of the thermal front. This would not be affected by the replenishment of the water bath with tap water during the experiment. Is this affected by the laboratory air? In that case, why was the increase smaller for Figure (a), which was conducted for a longer time?*

We will carefully check the experimental data, deduce an explanation and revise our manuscript accordingly.

*Interpretation of inverse peaks*

*I am glad to see Figure 5, illustrating the difficulty of the experiments. In my unpublished data, I observed similar inverse peaks for smaller grains (dp = 3 mm). I attributed this to the placement of fluid temperature sensors. It is extremely hard to make sure the depth of solid and fluid temperature measurement is the same. Smaller the grains are, more difficult. When I failed to do this, I observed inverse peaks regardless of fluid flow rates. Non-uniform flow could also cause the inverse peaks, but I do not think we can eliminate the possibility of misplacement of fluid temperature sensors relative to the location of the solid temperature sensors in this experimental setup.*

To fix the measurement position of the fluid and solid phase, a PVC frame was used as it is shown in Fig.1c. Although the LTNE probes were carefully embedded into the porous media composed by small glass beads, the possibility of small misplacement is inevitable. This influence could be evaluated by checking the same temperature measurement pair for fluid and solid phase, for example if they always produce the same inverse pulse. Considering this

comment, the manuscript will be revised including an improved explanation of the possible influence by any inaccuracy in sensor placement.

*Minor points*

*41: "at the same temperature at their interface": I believe this is true for LTNE approaches. The LTE approach assumes the temperature of the phases are the same within an REV, not just their interface.*

Thank for your commenting on the LTE definition. As you suggested the LTE model could be described by stressing the difference from the LTNE model. This sentence will be updated on the revised manuscript.

*99: Could you provide the information on the glue used (e.g., the name of the product)? Also, what is the volumetric heat capacity of the glue?*

Thermal Bonding System TBS20S, Electrolube was used, which is a two-part (Part A & Part B) thermally conductive epoxy system. Two parts were mixed with a ratio by volume (A:B) 3:1. According to the technical data sheet of the manufacture, specific heat capacity of Part A and Part B is 0.5 cal/g/°C at 30 °C and 0.35 cal/g/°C at 30 °C, respectively. And the cured glue density is 1.85 g/ml. We will add this to our revised manuscript.

*100: How did you place temperature sensors next to the surface of the glass spheres? Accurately placing sensors for fluid temperatures is important to avoid the "inverse peaks".*

PVC frames were designed to place the fluid and solid phase temperature sensors at the accurate positions. Through holes at the frame, PT100 sensor cables were inserted to be fixed aligning in a line. Please find a photo attached below the reply. And then the frame was placed at the depth of measurement inside the column, while the column was filled with small glass beads.

We believe that inverse peaks are a real phenomenon that cannot be avoided entirely. However, as mentioned previously, we will try to disentangle the effects of non-uniform flow from sensor placement by having a closer look at the dataset.

[Figure]

*111: Could you describe how you achieved water saturation of the porous media? Also, did you use any thermal insulation for the column? Minimizing air in porous media and heat loss from the column is essential to get experimental data that is compatible with the LTE model.*

Porous media was filled slowly with water from bottom to the top to avoid air bubble inside. The column was covered by an insulation layer. However, the laboratory was not equipped with climate control. Therefore, taking the air temperature into account, the initial water temperature was established close to the room temperature to minimize heat loss to the air. We will add this information to the revised manuscript.

*131: 26-34: "C" is missing.*

This will be corrected on the revised manuscript.

*Equation 1: "x is spatial coordinate" is missing.*

The definition of "x" will be added for the Eq. 1 on the manuscript.

*Equation 6: This analytical solution is for normalized temperature, not actual temperature T.*

The definition of T, T1 and T0 will be corrected.

*183: The effective thermal conductivity of fluid includes the effect of thermal dispersion under advection. Bandai et al., 2023 was not accurate for this description. Line 206 is accurate.*

Thank you for the specific comments. Line 180-182 will be revised considering the accurate definition of effective thermal dispersion.

*200: 1.5 m (= L)?*

Thank you for spotting the typo which will be corrected.

*Figure 2: I would suggest using the same color for the temperatures measured at the same depths for both phases.*

Thank you for your suggestion. We will consider the descriptive colour legend which can show intuitive colour code for measurement results from LTNE probes.

*Figure 8: There is a discrepancy between the models and the data at the end of the thermal front (in addition to the end of the breakthrough curve) for q = 17.2 m d-1. What caused this discrepancy? Heat loss from the column can be one reason, but if this was the case, the discrepancy would have been larger for dp = 25 mm, which was located at a deeper depth. Or, the location of the sensors for dp = 5 mm was not far enough from the input? But, if this was the case, the discrepancy would have been larger for q = 22.8 m d-1. Another reason may be the artifact of the temperature calibration procedure.*

We will conduct further data analysis to answer this question by thoroughly evaluating the cause of discrepancy between experimental data and model.

*344: "limited to 0.04 K": This is not accurate. This value is maximum normalized temperature difference in Figure 8 in Bandai et al., 2023, which can be converted to approximately 0.6 K because the temperature difference was about 15 K.*

Thank you for correcting the misunderstanding. This sentence will be revised considering the fact you stated.

*Line 379: Could you describe how you fitted the models to the experimental data to estimate the heat transfer parameters? It would be better to define a minimization problem to be solved and specify optimization algorithms used to solve it.*

We will evaluate the RMSE for each model (as was also raised by other reviewers) in our revised manuscript.